# Second Victims among German Emergency Medical Services Physicians (SeViD-III-Study)

**DOI:** 10.3390/ijerph20054267

**Published:** 2023-02-28

**Authors:** Hartwig Marung, Reinhard Strametz, Hannah Roesner, Florian Reifferscheid, Rainer Petzina, Victoria Klemm, Milena Trifunovic-Koenig, Stefan Bushuven

**Affiliations:** 1Department Health Sciences, MSH Medical School Hamburg, 20457 Hamburg, Germany; 2Wiesbaden Business School, RheinMain University of Applied Sciences, 65183 Wiesbaden, Germany; 3Department of Anaesthesiology and Intensive Care, Universitätsklinikum Schleswig-Holstein, Campus Kiel, 24105 Kiel, Germany; 4Bundesvereinigung der Arbeitsgemeinschaften der Notaerzte Deutschlands (BAND), 10969 Berlin, Germany; 5Training Center for Emergency Medicine (NOTIS e.V), 78234 Engen, Germany; 6Institute for Infection Control and Infection Prevention, Hegau-Jugendwerk Gailingen, Health Care Association District of Constance, 78315 Radolfzell, Germany; 7Institute for Medical Education, University Hospital, LMU Munich, 80539 Munich, Germany; 8Department of Anesthesiology and Critical Care, Medical Center—University of Freiburg, Faculty of Medicine, University of Freiburg, 79106 Freiburg im Breisgau, Germany

**Keywords:** second victim, medical error, emergency medical services, support strategies, coping, patient safety

## Abstract

Background: Patient care in the prehospital emergency setting is error-prone. Wu’s publications on the second victim syndrome made very clear that medical errors may lead to severe emotional injury on the caregiver’s part. So far, little is known about the extent of the problem within the field of prehospital emergency care. Our study aimed at identifying the prevalence of the Second Victim Phenomenon among Emergency Medical Services (EMS) physicians in Germany. Methods: Web-based distribution of the SeViD questionnaire among n = 12.000 members of the German Prehospital Emergency Physician Association (BAND) to assess general experience, symptoms and support strategies associated with the Second Victim Phenomenon. Results: In total, 401 participants fully completed the survey, 69.1% were male and the majority (91.2%) were board-certified in prehospital emergency medicine. The median length of experience in this field of medicine was 11 years. Out of 401 participants, 213 (53.1%) had experienced at least one second victim incident. Self-perceived time to full recovery was up to one month according to 57.7% (123) and more than one month to 31.0% (66) of the participants. A total of 11.3% (24) had not fully recovered by the time of the survey. Overall, 12-month prevalence was 13.7% (55/401). The COVID-19 pandemic had little effect on SVP prevalence within this specific sample. Conclusions: Our data indicate that the Second Victim Phenomenon is very frequent among prehospital emergency physicians in Germany. However, four out of ten caregivers affected did not seek or receive any assistance in coping with this stressful situation. One out of nine respondents had not yet fully recovered by the time of the survey. Effective support networks, e.g., easy access to psychological and legal counseling as well as the opportunity to discuss ethical issues, are urgently required in order to prevent employees from further harm, to keep healthcare professionals from leaving this field of medical care and to maintain a high level of system safety and well-being of subsequent patients.

## 1. Introduction

Prehospital emergency care is often performed under adverse conditions as time constraints, lack of information or a hazardous environment, making this subspecialty error-prone. Since Wu’s pioneering work, it has been a well-known fact that blame and isolation in view of medical errors may lead to emotional injury as represented by the so-called “Second Victim Phenomenon” (SVP) [1]. According to the most recently published consented definition a Second Victim (SV) is “any health care worker, directly or indirectly involved in an unanticipated adverse patient event, unintentional healthcare error, or patient injury and who becomes victimized in the sense that they are also negatively impacted” [2]. SVP is often triggered by unanticipated adverse events and may result in negative employee-related outcomes ranging from impaired quality of life to post-traumatic stress disorder (PTSD) and in some cases eventually suicide [3,4,5]. In addition, the care of future patients may be negatively affected [6]. Previous surveys in English-speaking countries suggest a prevalence up to 42% of second victims among healthcare professionals [7,8]. To face this challenge, interventional programs for healthcare professionals have been launched, mostly in English speaking countries and have improved employee-related outcomes [9]. To this day, however, there is a paucity of information and knowledge about the severity of this problem in professionals within this specific field of care [10]. For emergency services in Germany, where physician involvement in out-of-hospital emergencies has been widely established since the second half of the 20th century, this is even more true. Only recently, a first-time evaluation of this issue has been performed within a sample of residents in internal medicine at German acute care facilities (SeViD-I) and among acute care nurses (SeViD-II) [11,12]. Our study aimed at identifying the prevalence and characteristics of the Second Victim Phenomenon among doctors staffing EMS ambulances and helicopters in Germany. The primary goals of this survey were to assess prevalence and characteristics concerning SVP among pre-hospital EMS physicians to develop support strategies to facilitate coping with the consequences of medical error and thus to maintain a high level of patient and employee safety in prehospital emergency medicine. The secondary aim was to explore potential risk factors of having a high symptom load or prolonged recovery.

## 2. Materials and Methods

### 2.1. Construction and Validation of the SeViD Questionnaire

The development and validation of the questionnaire is described in detail in another publication [13] and has been referred to in the SeViD-I and -II publications. It is based on three domains: general experience with the Second Victim Phenomenon, prevalence of second victim symptoms and second victim support strategies and is using 40 items. BFI-10 was used to assess the personality traits of participants, including openness, neuroticism, agreeableness, extraversion and conscientiousness [14].

### 2.2. Design and Conduction of the SeViD-III Survey

The survey was conducted via web-based distribution of the SeViD questionnaire using the SurveyMonkey platform among the 12 regional organisations constituting the German Prehospital Emergency Physician Association (Bundesvereinigung der Arbeitsgemeinschaften der Notaerzte Deutschlands, BAND) representing approximately 12,000 members. The eight-week study period was from 12 September to 8 November 2022, and reminders were sent out after two, four and six weeks. Data collection was anonymized completely with neither tokens, cookies nor IP addresses stored. The invitation letter to participation as well as the reminders gave a short overview of the aims of the study, a brief description of the Second Victim Phenomenon and contained a QR code leading to the internet link. Ten items with regard to, among others, sociodemographic data, formal medical education, and work hours per week were added to the SeViD questionnaire. As the SARS-CoV-2 pandemic may have favoured incidents involving medical error we added an item to illustrate whether key incidents had been associated with the pandemic.

The survey used adaptive questioning so that items from the symptoms domain were displayed only to participants indicating that they had experienced specific incidents. Answering each question was mandatory to proceed to the next item. The participants were able to move backwards to questions they had answered already and to use a commentary function to give their suggestions at the end of the survey.

### 2.3. Measurements, Preparation and Re-Coding of Variables for Statistical Analysis

The continuous variables “age” and “work years” were categorized in three equally sized groups. The item “workplace” was dichotomized into acute care (intensive/intermediate care, emergency department, operation theatre) vs. “other”. Further dichotomizations were carried out with regard to second victim status (having experienced one or several second victim incidents vs. never experienced such incidents) and time to self-perceived recovery after the key incident (one month or less vs. more than one month). To assess symptom load sum scores were determined. Answer option “strongly pronounced” was accounted as “1”, the option “weakly pronounced” as “0.5” and “not at all” and “I don’t know” as “0”. 

### 2.4. Statistical Analysis

Expected and observed distribution was compared using contingency tables and analyzed for statistical significance by using chi^2^ testing. The influence of independent variables such as gender, age, professional experience, workplace in acute care and personality dimensions on the dichotomous dependent variable, second victim experienced (yes vs. no), were assessed using binary logistic regression models with bootstrapping, bias-corrected and accelerated (BCa) method based on 1000 samples. The influence of predictor variables sex, age, professional experience, workplace and personality traits on symptom load was tested with multiple linear regression with bootstrapping, the BCa method, based on 1000 samples. If multicollinearity was present, we centered (weighted by the mean values) the predictor variables prior to conducting regression analyses [15]. We added each predictor in the model in a separate step starting from the demographic variables, professional experience and workplace up to personality traits. Predictors, with decreased magnitude of effect size after a new predictor was included in the equation, were tested for possible mediating effects using PROCESS macro for SPSS v3.4 [16]. The significance level was set at α = 0.05. Statistical analysis was performed using SPSS Statistics Version 29 (IBM, New York, NY, USA).

## 3. Results

### 3.1. Baseline Characteristics

Out of 447 participants, 401 (90%) fully completed the survey. The mean time for participation was 6 min and 41 s. The median age was 43 (range 26 to 74); 69.1% were male. A total of 91.2% were board-certified in prehospital emergency medicine (“Zusatzbezeichnung Notfallmedizin”). The median experience in this field of medicine was 11 years with a minimum of one year and a maximum of 40 years. The mean working time in patient care during the last 12 months was 10.4 (median 12) months. In addition to their occupation in prehospital emergency care, 55.9% of all respondents were employed in operating rooms, 46.1% in intensive care units and 22.9% in emergency departments (multiple answers were accepted). Details of baseline characteristics are given in Table 1.

### 3.2. Second Victim Status

Statistical analysis suggests that 53.1% (213/401) had experienced a second victim incident once (24.9%) or on more than one occasion (28.2%) and that 25.8% (55/213) of second victims had experienced at least one incident in the 12 months preceding the survey resulting in an overall one year-prevalence of 13.7% (55/401). Adverse events resulting in patient harm (29.6%) and unexpected patient deaths or suicides (28.6%) were most highly ranked as key events. Out of the 213 persons affected by the SVP phenomenon, 48.8% (104) had received third party support from colleagues and 44.6% (95) from their families and friends while dealing with the incident. Further sources of support were superiors (16.0%/34), health care professionals (psychiatrists, psychologists; 13.6%/29) and, in one case, administration staff, while 5.6% (12) did not receive any support despite asking for it and a total of 35.2% (75) did not seek any form of support. Self-perceived time to full recovery was up to one month according to 57.7% (123) and more than one month for 31% (66) of the participants. In total, 11.3% (24) had not fully recovered by the time of the survey. The incident was related to the COVID-19 pandemic in 4.7% (10) of all events. 

### 3.3. Risks Factors for Becoming a Second Victim

Out of the “Big Five” personality traits, i.e., openness, conscientiousness, extraversion, agreeableness and neuroticism, only the latter proved to be a risk factor for becoming a Second Victim according to logistic regression analysis (regression coefficient B = 0.37; BCa 95% CI [0.04, 0.70], Odds Ratio (exponentiation of the B coefficient (Exp(B)) = 1.44; 95% CI [1.10, 1.88]). The results of binary logistic regression for risk factors for becoming a second victim are shown in Table 2. 

### 3.4. Factors with Impact on Symptom Load 

Within this sample, participants’ age and sex did not prove to be predictors of a high symptom load. After the inclusion of professional experience in the regression model, the results showed that professional experience as an EMS physician was associated with SV status, i.e., the less experience, the higher the symptom load (unstandardized regression coefficient B = −1.97; *p* < 0.05; BCa 95% CI [−3.60, −0.42], see Table 3). We could not perform multiple regression with workplace in acute care (yes vs. no) since this variable was constant after listwise exclusion of missing values. After we added the five personality scales in the regression equation, the impact of professional experience as an EMS physician on symptom load was no longer significant. Thus, the only significant predictor was neuroticism in the final model (B = 0.91, BCa 95% CI [0.35, 1.48], see Table 4). Since professional experience decreased its magnitude of effect size after neuroticism had been added in the model, we assumed that neuroticism might have acted as a mediator variable in the relationship between professional experience and symptom load, i.e., the influence of professional experience on symptom load may take an indirect path through neuroticism as a mediator variable. Consequentially, we tested the indirect effect of experience as an EMS physician via neuroticism on symptom load (see Figure 1) using SPSS PROCESS macro, model 4 [17]. The model applies the bootstrapping method per default with the deviation correction at 95% confidence interval based on 5000 samples to estimate direct, indirect and total effects.

We performed the mediation analysis with professional experience as an idenpendent variable, neuroticism as a mediator and symptom load as a dependent variable. The results revealed a negative significant indirect effect of professional experience on symptom load b = −0.27, bootstrapped 95% CI [−0.54, −0.06]. Furthermore, the direct effect of professional experience on symptom load in the presence of the mediator neuroticism was found to be significant (b = −0.92; *p* = 0.01). Hence, neuroticism partially mediated the relationship between professional experience and symtom load caused by second victim experience. Mediation analysis summary is presented in Table 5. 

### 3.5. Support Strategies 

Assessment of support strategies for coping with a serious event did not differ significantly comparing the group of physicians having experienced SVP (Table 6, columns 3 and 4) versus the ones who had never gone through such crisis (columns 1 and 2). The only exception was concerning the question of immediate time out to recover which was less often favoured by the Second Victim group.

## 4. Discussion

Our study aimed at investigating the frequency and severity of the Second Victim Phenomenon (SVP) among prehospital emergency physicians in Germany. The primary goals were to develop support strategies, to facilitate coping with the consequences of medical error, and thus to maintain a high level of patient and employee safety in prehospital emergency medical care. The survey revealed a 53.1% prevalence of the SVP among the target group of EMS physicians, resembling the results from two earlier studies among German residents in internal medicine and nurses which showed a 60% and 59% prevalence, respectively [12,13]. These findings far exceed the numbers that Seys et al. reported in their 2013 review which ranged from 10.4% up to 43.3% [3]. The reason for this difference might be an increasing awareness concerning the impact of medical errors on healthcare professionals during the last ten years. This awareness has also shed some light on the fact that the SVP may affect young healthcare professionals as early as during their university education or residency [17]. These findings correspond with our own results showing that the amount of professional experience as an EMS physician did have an impact on symptom load, i.e., the less experience, the higher the symptom load. However, this impact was indirect. Professional experience and neuroticism were both significantly associated with symptom load, noting that professional experience as an EMS physician as well as symptom load share the common proportion of variance with neuroticism as one out of five personality traits. Hence, after controlling for personality traits, the impact of professional experience on symptom load was no longer significant. This indicates that physicians who tend to be more experienced will tendentially experience lower symptom load because they tend to be less neurotic. 

The results of the mediating analysis show how relevant the personality traits for explanation of the work-related health impairment process are. For example, research concerning the job demands-resources model highlights the positive associations between neuroticism and job demands, as well as the negative associations between neuroticism and job resources and their impacts on burnout and work engagement [18,19]. These findings are in line with previous research which indicated that professional experience could act as a job resource and reduce the experience of strain, but only in individuals with a low extent of neuroticism. 

As a result of deeper analysis, we found formal and informal assistance to be beneficial and leading to less severe symptom load. These findings call for structures and processes such as mentoring programs and formalized feedback to prevent this vulnerable group from unnecessary harm. Furthermore, we were able to identify the personality traits that should be considered for further evaluation and future projects. For example, persons with high neuroticism scores not only tended to have a higher symptom load, but they were also vulnerable to SVP (as in SEVID-II), affirming the need to sensitize and, in case of SV events, care specifically for persons inclined towards perfectionism. Within this sample, the COVID-19 pandemic did not seem to have a significant impact on SVP prevalence. Just recently a study group from the USA pointed out that there may be a link between physician burnout and SVP in quite a surprising way, namely, that SVP prevalence increased in doctors with no present burnout and a large extent of moral injury was predictable burnout during the pandemic [5]. The authors concluded that predictive factors—especially for burnout as one of the most common causes for physicians leaving practice, a growing trend the authors call an “exodus”—must be well understood to prevent future shortages in physician staffing. Those shortages are already a reality in many German EMS systems and every effort is necessary to stop qualified and experienced physicians from leaving this demanding field of care [6]. 

One out of three SV in our sample had never asked for support from colleagues or family. As shown in a recent qualitative study from the United States, effective support strategies are not easy to establish and require a large amount of time and effort [8]. The authors suggest that activities incorporating the SVP within quality improvement processes may enhance program success. Moreover, the benefits of systematic efforts towards better coping seem to undoubtedly outweigh the effort if the value of those programs is fully recognized and systematic safety enhancements are achieved. These conclusions coincide with the recent findings from a review of SVP support resources by Busch et al. [9]. The authors’ statement that *“investing in second victim support structures should be a top priority for healthcare institutions adopting a systemic approach to safety and striving for just culture”* is fully supported by us. Due to their analysis, however, SVP program implementation and long-term success were often impeded by a persistent blame culture, limited awareness of program availability among the medical staff and a shortage of financial resources. Additionally, a potential overconfidence bias to be able to continue work after an incident might be regarded as a relevant obstacle in implementing support programs at peer level, as also assessed in another recent study [20]. To gain sustainable improvement in error management and safety culture, all those limiting factors have to be addressed and ultimately resolved. As far as adequate action after severe incidents is concerned, Liukka and her expert group from Finland undertook a systematic review of the literature published between 2009 and 2018. The authors call for comprehensive damage prevention after an adverse event and that actions should be taken immediately. To achieve this goal, they emphasize that improvements at a systemic level are required [6]. A new approach to peer group intervention after adverse events, the so-called “Buddy Study Program” from Denmark initiated by Schrøder et al., has proven to encourage a more compassionate working environment, raise attention concerning the wellbeing of co-workers and create a safe space for sharing [9].

More than 70% of the respondents to this survey had a professional background in anesthesiology. At the time this article was prepared, it had been almost ten years since the German Association for Anesthesiology and Intensive Care Medicine (Deutsche Gesellschaft für Anaesthesiologie und Intensivmedizin, DGAI) published recommendations for dealing with severe complications within this similarly error-prone subspecialty of medicine. That recommendation highlights several steps that most of the expert teams quoted in this discussion have been calling for, such as peer support, legal advice, etc. However, the level of awareness for those recommendations does not seem to be high enough and it requires continuous effort to spread the word and gain systematic improvement [21]. The experience of SVP might be replicated for emergency physician trainees in Germany (NAsim-25) by innovative high-fidelity simulation-based programs, raising participants’ awareness of the issue of medical errors and the fact that patient safety is often at stake in the specific field of emergency care [22]. 

Our findings suggest that life-long medical education, team management and leadership programs should comprise knowledge and skills to be prepared for SVP, to recognize SVP and to seek further assistance in predisposing situations. Further expertise in “first psychological aid” formats should be evaluated. Considering the shortage of EMS personnel, expertise in SVE is demanded not only for providers themselves, but also for supervisors, other health care professionals and institutions.

Certain limitations to our findings have to be considered: First, we are not able to determine an exact response rate. The link to the survey was sent out by the region, comprising the federal organization, via e-mail, to the approximately 12,000 EMS physicians organized in Germany; therefore, we are unable to determine how many of those members were actually reached via e-mail and thus had access to the survey link. For example, a relevant number of e-mail addresses may have been either missing or outdated and e-mails may have been mismatched by spam filters. At the same time, we are quite sure that no participants from outside our desired study sample took part in the survey, as the study link was never available via social media or publicly accessible websites. Second, as with any cross-sectional design, the descriptive statistics are unable to identify cause-and-effect relationships. Further limitations exist and are described in more detail in the SeViD-I- and -II-studies. Still, research in this error-prone field of care seems worthwhile and should be pursued in the future. 

## 5. Conclusions

To our knowledge, this survey represents one of the first investigations of the Second Victim Phenomenon within the Emergency Medical Services (EMS) environment worldwide. Our data indicate that the Second Victim Phenomenon is frequent among prehospital emergency physicians. Formally established support networks, e.g., access to psychological and legal counseling, are urgently required in order to prevent employees from additional harm. To further evaluate the status quo within this specific field of medical care, we will perform additional surveys among emergency medical dispatchers and paramedics in the near future.

## Figures and Tables

**Figure 1 ijerph-20-04267-f001:**
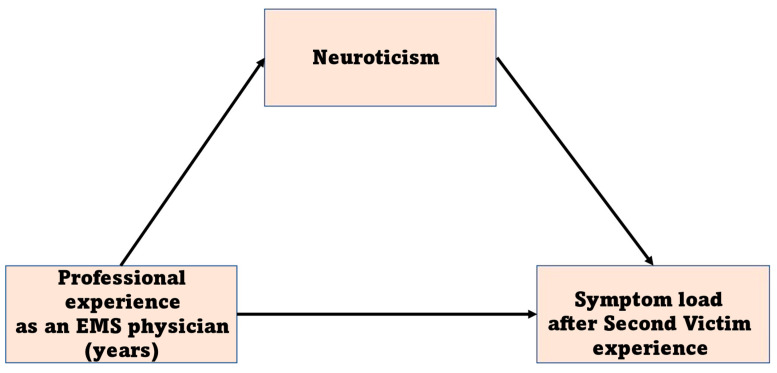
Mediation model. EMS: Emergency medical services, Neuroticism: One out of Big Five Personality Traits, Symptom load: the sum of symptoms after second victim experience.

**Table 1 ijerph-20-04267-t001:** Baseline Characteristics.

Total number of fully completed surveys		401
Gender (female/male/diverse)		30.9% (124)/69.1% (277)/-
Age (years)	≤39	35.7% (143)
	40–48	31.9% (128)
	49–74	32.4% (130)
Formal education *	Board-certified EMS physician (current)	91.2% (369)
	Board-certified anesthesiologist	63.6% (255)
	Certified EMS physician (obsolete)	27.2% (102)
	Board-certified clinical emergency physician	12.0% (48)
	Anesthesiology resident	9.7% (39)
	Board-certified internist	9.2% (37)
	Board-certified general surgeon/trauma surgeon	9.0% (36)
	Board-certified general practitioner	4.2% (17)
	Other	13.7% (55)
Professional experience as an EMS physician (years)	Median (min/max)	11 (1/40)
Leading position		51.4% (206)
Full vs. part time occupation		68.3 vs. 31.7%
Place of occupation	Operating Room	55.9% (224)
	Intensive Care/Intermediate Care	46.1% (185)
	Emergency Department	22.9% (92)
	Registered Practice	9.0% (36)
	General Ward	4.7 (19)
	Other	21.9% (88)
Working mode (time)	Irregular, no shift work	39.2% (157)
	Shift work including nights	31.9% (128)
	Regular, daytime only	13.5% (54)
	Shift work without nights	3.5% (14)
	Other	11.9% (48)
Months spent in patient care during past year	Mean ± SD	10.4 ± 4
Openness	Mean ± SD	3.32 ± 0.97
Conscientiousness	Mean ± SD	3.99 ± 0.83
Extraversion	Mean ± SD	3.27 ± 0.97
Agreeableness	Mean ± SD	3.26 ± 0.78
Neuroticism	Mean ± SD	2.35 ± 0.86

* multiple answers permitted.

**Table 2 ijerph-20-04267-t002:** Risk factors for becoming a Second Victim. Results of binary logistic regression.

Independent Variable	Regression Coefficient B with BCa 95% CI	*p*	Odds Ratio (Exponentiation of the B Coefficient (Exp(B))	Odds Ratio 95% CI Lower	Odds Ratio 95% CI Upper
Gender (female) ^1^	−0.08 BCa 95% CI [−0.55, 0.43]	0.75	0.93	0.58	1.49
Age group ^2^ ≤ 39 40–48	0.16 BCa 95% CI [−0.60, 0.86] 0.83 BCa 95% CI [−0.80, 1.03]	0.60 0.86	1.167 1.09	0.66 0.46	1.17 1.09
Professional experience as an EMS physician (years)	−0.10 BCa 95% CI [−0.05, 0.03]	0.61	0.99	0.95	1.03
Workplace in acute care ^3^	−0.07 BCa 95% CI [−0.59, 0.43]	0.78	0.93	0.57	1.53
Openness to experience	0.14 BCa 95% CI [−0.11, 0.42]	0.22	1.15	0.92	1.44
Conscientiousness	0.03 BCa 95% CI [−0.26, 0.34]	0.84	1.03	0.79	1.35
Extraversion	−0.13 BCa 95% CI [−0.34, 0.05]	0.22	0.88	0.71	1.09
Agreeableness	0.30 BCa 95% CI [−0.01, 0.60]	0.03	1.35	1.03	1.78
Neuroticism	0.37 BCa 95% CI [0.04, 0.70]	0.01	1.44	1.10	1.88

Dependent variable is second victim experience (dichotomous yes vs. no), ^1^ referent category is male, ^2^ referent category is the age group 49–74, ^3^ referent category is emergency room, intensive care unit or operating theatre, EMS: Emergency medical services, BCa 95% CI: Bias-corrected and accelerated bootstrapping 95% confidence intervals based on 1000 samples.

**Table 3 ijerph-20-04267-t003:** Impact of demographic variables and professional experience as an EMS physician on symptom load. Results of multiple linear regression.

Independent Variable	Unstandardized Regression Coefficient B	*p*	BCa 95% CI Lower	BCa 95% CI Upper
Constant	3.59	0.11	−1.56	8.25
Gender (female = 1, male = 2)	−0.70	0.09	−1.87	0.20
Age ^1^	2.37	0.25	−1.73	9.81
Professional experience as an EMS physician (years) ^1^	−1.97	0.01	−3.60	−0.42

Dependent variable is symptom load caused by second victim experience. ^1^ variables are centred (weighted by mean values). EMS: Emergency medical services. Lower BCa 95% CI and Upper BCa 95% CI: lower and upper limits of 95% bias-corrected and accelerated bootstrapped confidence interval for unstandardized regression coefficient B.

**Table 4 ijerph-20-04267-t004:** Factors with impact on symptom load. Results of multiple linear regression.

Independent Variable	Unstandardized Regression Coefficient B	*p*	BCa 95% CI Lower	BCA 95% CI Upper
Constant	4.96	0.14	−1.46	11.49
Gender (female = 1, male = 2)	−0.70	0.18	−1.71	0.28
Age ^1^	2.37	0.43	−3.36	8.82
Professional experience as an EMS physician (years) ^1^	−1.57	0.05	−3.29	−0.01
Openness to experience	0.01	0.99	−0.55	0.51
Conscientiousness	−0.02	0.96	−0.56	0.53
Extraversion	−0.49	0.04	−0.91	−0.08
Agreeableness	0.22	0.52	−0.38	0.86
Neuroticism	0.91	0.00	0.35	1.48

Dependent variable is symptom load caused by second victim experience. ^1^ Independent variables are centred (weighted by mean values), EMS: Emergency medical services, Lower BCa 95% CI and Upper BCa 95% CI: lower and upper limits of 95 % bias-corrected and accelerated bootstrapped confidence interval of unstandardized regression coefficient B.

**Table 5 ijerph-20-04267-t005:** Indirect, direct, and total effects of length of professional experience as an EMS physician in years on symptom load caused by second victim experience.

Relationship	Total Effect	Direct Effect	Indirect Effect	95% CI of Indirect Effect [bootLLCI, bootULCI]	Conclusion
Professional experience < Neuroticism < Symptom load	−1.19	−0.92	−0.27	[−0.54, −0.06]	Partial mediation

EMS: Emergency medical services, 95% CI of indirect effect [bootLLCI, bootULCI]: lower and upper limits of 95% confidence interval based on 5000 deviation correction bootstrapped samples.

**Table 6 ijerph-20-04267-t006:** Ratings of Support Strategies subject to Second Victim (SV) Status.

Support Strategy	Rated Rather or very Helpful % (n) * No SV Status (n = 188)	Rated Rather Not or Not Helpful % (n) No SV Status (n = 188)	Rated Rather or very Helpful % (n) SV Status Present (n = 213)	Rated Rather Not or Not Helpful % (n) SV Status Present (n = 213)	*p* (chi2)
1. Immediate time out to recover	70.7 (133)	16.5 (31)	64.3 (137)	30.5 (65)	*p* = 0.001
2. Access to counseling including psychological/psychiatric services	87.8 (165)	7.4 (14)	83.1 (177)	11.7 (25)	*p* = 0.14
3. Opportunity to discuss emotional and ethical issues	94.7 (178)	2.1 (4)	93.9 (200)	4.7 (10)	*p* = 0.17
4. Concise and prompt information about procedures (e.g., root cause analysis, reporting)	86.7 (163)	10.1 (19)	88.3 (188)	8.9 (19)	*p* = 0.67
5. Formal peer support	84.0 (158)	10.1 (19)	84.0 (179)	12.7 (27)	*p* = 0.48
6. Informal emotional support	75.7 (140)	17.6 (33)	81.2 (173)	11.3 (24)	*p* = 0.07
7. Prompt debriefing/crisis intervention	91.0 (171)	5.3 (10)	90.6 (193)	7.0 (15)	*p* = 0.50
8. Supportive guidance for continuing professional duties	62.8 (118)	30.3 (57)	66.7 (142)	25.8 (55)	*p* = 0.29
9. Support for communicating with patients or relatives	71.8 (135)	24.5 (46)	64.8 (138)	29.1 (62)	*p* = 0.23
10. Specific regulations concerning professional conduct	59.0 (111)	34.0 (64)	57.3 (122)	29.6 (63)	*p* = 0.59
11. Support during active follow up of the incident	83.5 (157)	11.7 (22)	82.6 (176)	11.3 (24)	*p* = 0.95
12. Safe opportunity to contribute insights in order to prevent similar events in the future	83.5 (157)	13.3 (25)	85.9 (183)	7.5 (16)	*p* = 0.07
13. Access to legal counseling after severe events	94.7 (187)	3.7 (7)	88.7 (189)	5.6 (12)	*p* = 0.32

* Missing values to 100% are due to the answer option “I cannot judge this”.

## Data Availability

The data presented in this study are available upon request from the corresponding author.

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
