# Peer review of "Second Victims among German Emergency Medical Services Physicians (SeViD-III-Study)"

_ijerph, 2023, doi:10.3390/ijerph20054267_

Round 1
Reviewer 1 Report
Thanks for the opportunity to review this manuscript by Hartwig Marung et al. titled Second Victims Among German Emergency Medical Services 2 Physicians (SeViD-III-Study)
It is a very interetsing topic and study
Follwing some suggestion to improve the reading:
1 the intro is too short, the second victim fenomenon needs to be better explained and summarized before getting into the details of the study done
2 please do not start sentences with a number
3 table 2 does not add any particular information to the manuscript, it can be deleted or added to the supplements. The most important info with satitistical significance of teh table can be described in the text
3 discussion: is not too strong in highlighting the severity of the problem. For instance: what are the other reasons for this rise in the percentage of SVF apart from increasing awareness and higher number of young physicinas involved? Given the time of the survey after the COVID pandemics period I wonder if this might have contributed in any way…. Please articulate further hypotesis as this is the real core of the study
4. discussion: can specific medical education programs have a place in improving the SVF problem? Please discuss further
Thanks for this oportunuity and I look further to revieweing the revised manuscript
Author Response
Dear Reviewer,
we appreciate your time and effort to provide feedback on the first draft of our article. We have taken up your comments as follows:
Comments and Suggestions for Authors
Thanks for the opportunity to review this manuscript by Hartwig Marung et al. titled Second Victims Among German Emergency Medical Services 2 Physicians (SeViD-III-Study).
It is a very interesting topic and study.
Following some suggestion to improve the reading:
1 the intro is too short; the second victim phenomenon needs to be better explained and summarized before getting into the details of the study done.
We added some information explaining the Second Victim Phenomenon in more detail (lines 53-70) incorporating the recently published evidence-based definition of SV published by Vanhaecht et al. which perfectly explains the nature of SV to readers unfamiliar with this concept.
2 please do not start sentences with a number.
We have revised the manuscript accordingly.
3 table 2 does not add any particular information to the manuscript, it can be deleted or added to the supplements. The most important info with statistical significance of the table can be described in the text.
We have added a description of SV status to make our intentions clearer to readers. As we are able to show significant differences for one important aspect of peer support that might be associated with overconfidence limiting the effect of peer support, we still think that this additional information is very useful for readers. We added a detailed explanation about the implications of this deviation in the discussion starting at line 238.
3 discussion: is not too strong in highlighting the severity of the problem. For instance: what are the other reasons for this rise in the percentage of SVF apart from increasing awareness and higher number of young physicians involved? Given the time of the survey after the COVID pandemics period I wonder if this might have contributed in any way….
Based on our findings, the impact of the COVID pandemic was low (line 155) in accordance to other findings from SeViD-II and for this reason we did not elaborate this issue in more detail (line 212). However we agree about the severity of this problem, which to quote the inventor of the term Second Victim, Albert Wu, “is as old as medicine itself.”
Please articulate further hypothesis as this is the real core of the study.
We added further current evidence and hypothesis from researchers in this field to make our points more clear.
- discussion: can specific medical education programs have a place in improving the SVF problem? Please discuss further.
We integrated a reference to the so called “NAsim-25” high-fidelity simulation-based program for prehospital emergency physician trainees beginning with line 263.
Thanks for this opportunity and I look further to reviewing the revised manuscript.
Once again, we thank you both very much for your valuable support. We gratefully remain at your disposal for any further comments.
Best regards, for the authors
Hartwig Marung
Reviewer 2 Report
Dear authors, I read with interest your manuscript and I think that the study has been well conducted and properly described. I just ask you some details that I think are needed for a full evaluation of your work.
- Lines 130-132; section 3.2: the logistic regression analysis output is not clear. A p-val <0.01 is reported but no crude OR. Moreover, a 95% CI including the null value (1) is shown. This is inconsistent with what is declared. Please explain.
- Lines 133-137: same issue with section 3.3
- A section Data analysis is missing, where it should be explained the statistical plan and the software used.
- Table 2: From the text (lines 139-143), It seems that a comparison is made between NO SV status and SV Status. However, I think it is necessary to clearly identify at the top of the table the column related to the SV status. Moreover, it is not clear the statistical analysis conducted (and without a specific section there is no clue): did the researched compare all four groups (1-No SV rated rather or very helpful; 2-No SV rated rather not or not helpful; 3-SV rated rather or very helpful; 4- SV rated rather not or not helpful), or did they compare 1 to 3 and 3 to 4? Apparently, it is the first one because the second would have yielded two p values. Yet, maybe 1+2 vs 3+4 has been conducted. Again, it is necessary a data analysis section.
- I think it should be briefly discussed the fact that the questionnaire was sent to 12,000 professionals (if I understood well) but only 447 returned. Had these individuals been more sensible to the SV topic, the prevalence may be overstated. Even though it is correctly reported in the limitations, I am interested in the authors opinion about how this may or may not have affected their results.
Kind regards,
Author Response
Dear Reviewer,
we appreciate your time and effort to provide feedback on the first draft of our article. We have taken up your comments as follows:
- Lines 130-132; section 3.2: the logistic regression analysis output is not clear. A p-val <0.01 is reported but no crude OR. Moreover, a 95% CI including the null value (1) is shown. This is inconsistent with what is declared. Please explain.
We would like to apologize for this confusion. We made clerical errors in reporting the findings of the analyses. We conducted binary logistical regression for the dichotomous dependent variable Second Victim experience yes vs. To ensure robust regression coefficients, we applied bootstrapping (bias accelerated method, based on 1,000 samples). In revised version we report on Odds with 95%CI and B with BCa 95% CI. In contrast, we performed multiple linear regression with symptom load as an independent variable also with applied bootstrapping (bias accelerated method, based on 1,000 samples). We report these findings in table format (tables 2 and 3).
- Lines 133-137: same issue with section 3.3
- A section Data analysis is missing, where it should be explained the statistical plan and the software used.
We added section 2.4 (lines 116-126).
- Table 2: From the text (lines 139-143), It seems that a comparison is made between NO SV status and SV Status. However, I think it is necessary to clearly identify at the top of the table the column related to the SV status. Moreover, it is not clear the statistical analysis conducted (and without a specific section there is no clue): did the researched compare all four groups (1-No SV rated rather or very helpful; 2-No SV rated rather not or not helpful; 3-SV rated rather or very helpful; 4- SV rated rather not or not helpful), or did they compare 1 to 3 and 3 to 4? Apparently, it is the first one because the second would have yielded two p values. Yet, maybe 1+2 vs 3+4 has been conducted. Again, it is necessary a data analysis section.
We compared 1+2 to 3+4 and have explained this in more detail (line 184)
- I think it should be briefly discussed the fact that the questionnaire was sent to 12,000 professionals (if I understood well) but only 447 returned. Had these individuals been more sensible to the SV topic, the prevalence may be overstated. Even though it is correctly reported in the limitations, I am interested in the authors opinion about how this may or may not have affected their results.
We have explained this fact in more detail (lines 275-282).
Once again, we thank you both very much for your valuable support. We gratefully remain at your disposal for any further comments.
Best regards, for the authors
Hartwig Marung